# Abscisic Acid, Paclobutrazol, and Salicylic Acid Alleviate Salt Stress in *Populus talassica × Populus euphratica* by Modulating Plant Root Architecture, Photosynthesis, and the Antioxidant Defense System

**Mengxu Su** [1,2,†], **Min Zhang** [1,2,†], **Ying Liu** [1,2] and **Zhanjiang Han** [1,2,*]

1 College of Life Science and Technology, Tarim University, Alar 843300, China
2 Xinjiang Production and Construction Corps Key Laboratory of Protection and Utilization of Biological Resources in Tarim Basin, Alar 843300, China
* Correspondence: hanzhanjiang@taru.edu.cn
† These authors contributed equally to this work.

**Abstract:** The exogenous plant growth regulators (PGRs) represent a useful strategy for reducing the adverse effects of salt stress in plants. In order to investigate the regulatory effect of exogenous PGRs on the salt tolerance of *Populus talassica × Populus euphratica* seedlings, in this study, the effects of different foliar spray concentrations of ABA (5 mg·L$^{-1}$, 15 mg·L$^{-1}$, 25 mg·L$^{-1}$), PP$_{333}$ (300 mg·L$^{-1}$, 900 mg·L$^{-1}$, 1500 mg·L$^{-1}$), and SA (40 mg·L$^{-1}$, 120 mg·L$^{-1}$, 200 mg·L$^{-1}$) on *P. talassica × P. euphratica* seedlings under salt stress (soil salt concentration of 2%) were determined. The results showed that the dry weight, total root length, root surface area, root volume, total Chl content, and photosynthetic parameters of *P. talassica × P. euphratica* seedlings significantly decreased under salt stress and increased their contents of malondialdehyde (MDA), hydrogen peroxide (H$_2$O$_2$), superoxide anion (O$_2^-$), free proline (Pro), superoxide dismutase (SOD), and peroxidase (POD). However, ABA, PP$_{333}$, and SA can mitigate the adverse effects of salt stress on these indicators. Especially, the 15 mg·L$^{-1}$ ABA, 900 mg·L$^{-1}$ PP$_{333}$, and 120 mg·L$^{-1}$ SA treatments had the best effect on alleviating salt stress, with significant increases in dry weight, root parameters, total Chl content, and the photosynthetic parameters of the *P. talassica × P. euphratica* seedlings, improving their photosynthetic characteristics; meanwhile, increased Pro content and enzyme activity and decreased MDA, H$_2$O$_2$, and O$_2^-$ content protected the integrity of membrane system and enhanced the salt tolerance of the seedlings. SA resulted in a better ameliorative effect on salt stress compared to ABA and PP$_{333}$.

**Keywords:** plant growth regulators; foliar spray; soil salt concentration; salt tolerance





## 1. Introduction

The worsening of soil salinization is one of the main factors that restrict the growth and development of plants, endangers the ecological environment, and affects the sustainable development of the global agricultural and forestry economy and ecosystems. Soil salinization is a global environmental problem. The land worldwide that contains saline-alkali soils exceeds $8.3 \times 10^8$ hm$^2$, including 53% alkaline soils and 47% saline soils [1]. The problem is particularly severe in semiarid and arid areas and is predicted to be more drastic in the future [2]. It is estimated that the area of saline-alkali soil is growing at a rate of $1 \times 10^7$ to $1.5 \times 10^7$ hm$^2$ annually worldwide, resulting in serious salinization hazards and water and soil loss [3]. Countries seriously affected by saline soil mainly include Australia, Thailand, Syria, the United States, and China [4]. According to data from previous work, the area of saline soil in China is about $3.6 \times 10^7$ hm$^2$, accounting for 4.88% of the available land area in the country [5]. Xinjiang, located in the arid and semiarid areas of Northwest China, contains the largest area of salinized soil in China [6], which severely inhibits plant growth

and hinders sustainable forestry development. Therefore, the development and utilization of saline-alkali land is the primary task for the development of agricultural and forestry production. It is particularly important to vigorously promote salt-tolerant tree species and improve the adaptability of plants grown on saline-alkali land.

In general, excessive soil salt accumulation produces osmotic stress and specific ion toxicity in plants, which leads to secondary oxidative stress, membrane permeability, and plant death in severe cases [7]. It impairs all aspects of plant growth and development, manifesting as the inhibition of plant rooting, photosynthesis, physiology, biochemistry, and other aspects, which further adversely affect plant growth [8]. Maintaining a high cytosol $K^+/Na^+$ ratio can mitigate stress-induced deleterious changes, which is an important aspect of maintaining physiological cellular functions [9]. However, under salt stress, a large amount of $Na^+$ and $Cl^-$ increases the osmotic pressure of soil water, causing the osmotic pressure in plants to be lower than that in the soil. Meanwhile, because of the antagonistic relationship with $Na^+$, the high accumulation of $Na^+$ also inhibits the uptake of $K^+$, $Ca^{2+}$, and P by plants [10,11]. Therefore, the uptake of large amounts of salt ions by plant roots can disrupt the original $K^+/Na^+$ balance in the cell, disrupting the plant's ability to maintain the ionic balance inside and outside the cell, and ultimately affecting the plant's entire metabolism. As the first plant organ under stress [12], when the water absorption capacity of roots becomes greatly inhibited, and physiological drought occurs, the plant root architecture changes accordingly [13], thereby slowing plant growth rate. When a large amount of $Na^+$ is absorbed by plants and accumulates in the body, it destroys cell membranes, hinders biosynthesis in the cytoplasm, and causes lipid peroxidation damage [14]. Moreover, it inhibits the activity of photosynthesis-related enzymes, limits the photosynthetic rate, and damages the photosynthetic system [15].

In recent years, plant growth regulators have been used to alleviate salt-stress-induced plant damage, and their relationship with plant salt tolerance has been widely studied. Abscisic acid is a small sesquiterpene molecule and has an important regulatory function on plant growth and development and stress resistance, which is a common plant growth regulator [16]. Under salt stress, ABA can effectively improve the antioxidant capacity of tomato seedlings, reduce the levels of reactive oxygen species (ROS) and malondialdehyde (MDA), increase the content of proline, and reduce stomatal conductance, which is conducive to promoting the growth of tomato seedlings [17]. Spraying ABA can adjust the levels of ions and organic solutes of wheat seedlings under salt stress, increasing their aboveground biomass and promoting growth [18]. $PP_{333}$ is a stress protectant with growth regulating properties, which is now extensively used in regulating plant development under normal or stressed conditions. It can promote plant growth, photosynthesis, and other functions by improving the level of osmolytes, antioxidant activity, maintaining membrane stability, and photosynthetic pigments [19,20]. $PP_{333}$ increases the fresh and dry weight of the shoots and roots of tomato plants and enhances Chl content, Pro content, and SOD activity, reducing the adverse effects of salt stress and enhancing the salt tolerance of their plants [21]. $PP_{333}$ has been shown to promote plant height, dry and fresh weight, and improve the antioxidant defense system of *Vigna unguiculata* [22] and *Catharanthus roseus* [23], which significantly ameliorated the adverse effects of salt stress. SA was first separated from willow bark by the German scientist Johann Buchner in 1828, and it has been shown to be an important hormone involved in regulating plant photosynthesis, the antioxidant defense system, and other physiological processes [24]. For example, the application of SA has been reported to markedly reduce the content of malondialdehyde (MDA) and increase the activity of antioxidant enzymes in *P. euphratica* grown under salt stress, alleviating oxidative damage to the plasma membrane [25]. The application of SA can also significantly improve osmotic potential, the Chl content, and photosynthetic capacity of tomato plants under salt stress while reducing the content of MDA and Pro, preventing membrane damage and promoting the accumulation of dry matter [26]. However, the effective concentration of ABA, $PP_{333}$, and SA for alleviating salt damage depends on multiple parameters. Especially in trees, the mechanism of abiotic stress response and tolerance of ABA, $PP_{333}$, and SA, still needs to be

investigated. Tomato seeds with 1 mM SA could enhance the seed vigor index, increase the germination rate, and increase the activities of CAT, POD, and SOD, thus reducing the adverse effects of salinity [27]. At the same time, it can also increase the germination rate of wheat seeds and seedling growth, increase the content of photosynthetic pigments, and improve salt tolerance [28].

*P. talassica* × *P. euphratica*, an excellent tree variety, is crossbred with *P. talassica* as the female parent and *P. euphratica* as the male parent by the Xinjiang Academy of Forestry Sciences and Jimusar County Forest Variety Test Station (Xinjiang, China) [29,30]. It has the characteristics of rapid growth, a strong asexual reproduction ability, and a high survival rate for cuttings. In addition, it has an excellent appearance and offers good landscaping material, with good stress resistance to salt alkali, cold, drought, wind, and sand. It is widely planted in the saline-alkali areas of Northwest China [31]. Our research team has made some progress in unraveling salt stress physiology and in undertaking salt tolerance-related gene mining in *P. talassica* × *P. euphratica* [29], but the salt tolerance mechanism of *P. talassica* × *P. euphratica* is very complex and needs further research. In order to study and improve the salt tolerance of *P. talassica* × *P. euphratica* and promote large-scale planting and promotion, the mitigating effects of ABA, PP$_{333}$, and SA on different plants under salt stress have been used as a reference in previous studies. In this study, the optimal concentrations of these exogenous plant growth regulators for *P. talassica* × *P. euphratica* seedlings under salt stress were investigated to clarify the regulatory mechanism of salt tolerance and provide a reference and theoretical basis for the use of plant growth regulators in alleviating growth inhibition in seedlings under salt stress.

## 2. Materials and Methods

### 2.1. Overview of the Study Site

The study site is located in the *P. talassica* × *P. euphratica* nursery (81°18′08″ E, 40°36′13″ N and altitude of 1014 m) of the seedling base of the 10th Regiment of the 1st Division of the Xinjiang Production and Construction Corps. The area has an extreme continental arid desert climate in a warm temperate zone. The average annual sunshine hours are 2556.3–2991.8 h; temperatures are 10–12 °C; the evaporation is 1876.6–2558.9 mm, with average annual precipitation of 40.1–82.5 mm. There is sufficient light, large temperature differences between day and night, scarce annual rainfall, and strong surface evaporation.

### 2.2. Plant Material and Experimental Treatments

Two-year-old potted *P. talassica* × *P. euphratica* seedlings were used to study the alleviating effects of different plant growth regulators on salt stress using open-air potted soil culture. Each pot (30 cm in upper diameter, 20 cm in lower diameter, and 25 cm in height) was filled with 8 kg of experimental soil with pH, organic matter, total nitrogen, alkaline nitrogen, available phosphorus, available potassium, Na$^+$, K$^+$, and Ca$^{2+}$ of 7.56, 26.55 g·kg$^{-1}$, 0.92 g·kg$^{-1}$, 22.11 mg·kg$^{-1}$, 24.02 mg·kg$^{-1}$, 124.68 mg·kg$^{-1}$, 0.14 mg·g$^{-1}$, 0.03 mg·g$^{-1}$, and 0.11 mg·g$^{-1}$, respectively. The initial soil salt content was 0.11%. Referring to previous research of this research group [30], the treatment groups were watered with 350 mmol·L$^{-1}$ NaCl solution once every 3 d, until the soil salt content (SSc) reached 2%. The CK group was irrigated with deionized water each time.

After salt treatment, ABA, PP$_{333}$, and SA were sprayed onto the front and back of the leaves every morning and evening for four consecutive days, with distilled water as a control. All seedlings were randomly divided into 11 groups of 6 plants each. The experimental design was as follows: (i) CK (control): foliar spray with water and no NaCl treatment; (ii) NaCl: foliar spray with deionized water and 2% SSc; (iii) A1: 5 mg·L$^{-1}$ ABA + 2% SSc; (iv) A2: 15 mg·L$^{-1}$ ABA + 2% SSc; (v) A3: 25 mg·L$^{-1}$ ABA + 2% SSc; (vi) P1: 300 mg·L$^{-1}$ PP$_{333}$ + 2% SSc; (vii) P2: 900 mg·L$^{-1}$ PP$_{333}$ + 2% SSc; (viii) P3: 1500 mg·L$^{-1}$ PP$_{333}$ + 2% SSc; (ix) S1: 40 mg·L$^{-1}$ SA + 2% SSc; (x) S2: 120 mg·L$^{-1}$ SA + 2% SSc; and (xi) S3: 200 mg·L$^{-1}$ SA + 2% SSc (Table 1).

**Table 1.** Settings for various treatment combinations.

| Treatment | Soil Salt Content(SSc) | Plant Growth Regulators(PGRs) |
|---|---|---|
| CK | 0.11% | - |
| NaCl | 2% | - |
| A1 | 2% | $5 \ mg \cdot L^{-1}$ ABA |
| A2 | 2% | $15 \ mg \cdot L^{-1}$ ABA |
| A3 | 2% | $25 \ mg \cdot L^{-1}$ ABA |
| P1 | 2% | $300 \ mg \cdot L^{-1}$ $PP_{333}$ |
| P2 | 2% | $900 \ mg \cdot L^{-1}$ $PP_{333}$ |
| P3 | 2% | $1500 \ mg \cdot L^{-1}$ $PP_{333}$ |
| S1 | 2% | $40 \ mg \cdot L^{-1}$ SA |
| S2 | 2% | $120 \ mg \cdot L^{-1}$ SA |
| S3 | 2% | $200 \ mg \cdot L^{-1}$ SA |

After 45 days of culture, all indices were determined to analyze dry mass, root structure parameters, Chl content, photosynthetic parameters, concentrations of MDA, $H_2O_2$, $O_2^-$, Pro, and antioxidant enzyme activity. The leaves used for the MDA, $H_2O_2$, $O_2^-$, Pro, and antioxidant enzyme activity analyses were frozen in liquid nitrogen after collection and stored at $-80 \ ^\circ C$. Each treatment contained three biological replicates.

### 2.3. Measurement of Growth Index

Three plants were randomly selected from each treatment and taken to the laboratory. After cleaning with deionized water, the seedlings were placed in Kraft paper bags. The samples were then dried in an electrothermal constant temperature drying oven at $105 \ ^\circ C$ for 15 min and at $75 \ ^\circ C$ until a constant mass was attained and weighed. Root images were obtained using a Microtek scanmaker i800 Plus (Shanghai Microtek Technology Co., Ltd., Shanghai, China) and total length, surface area, and volume were measured using a Wanshen LA-S series plant image analyzer system (Hangzhou Wanshen Testing Technology Co., Ltd., Hangzhou, China).

### 2.4. Measurements of Photosynthetic Parameters and Total Chlorophyll Content

The leaf photosynthetic parameters were measured from 11:00–13:00. Net photosynthetic rate (Pn), stomatal conductance (Gs), and transpiration rate (Tr) were measured under a photosynthetically active radiation (PAR) of $1200 \ \mu mol \cdot m^{-2} \cdot s^{-1}$ and a $CO_2$ concentration of $390 \ \mu mol \cdot m^{-2} \cdot s^{-1}$ at $25 \ ^\circ C$ using a Portable Photosynthesis System (LI-6400). Ten leaves from each sampling point were measured, with three replicates.

We took 0.05 g of fresh leaves, cut them into filaments and put the samples into 5 mL of 95% ethanol in the dark for 24 h. Then, the extracted solution was analyzed. We used 3 mL of extract to measure the Chl absorbance at 649 and 665 nm. The Chl contents ($mg \cdot g^{-1}$) were determined by applying the absorbance values to the equations reported by Lichtenthaler and Wellburn for ethanol [32]. Total Chl content was calculated as follows:

$$Chl \ mg \cdot g^{-1} \ FW = \frac{(6.63A_{665} + 18.08A_{649}) \times V}{W \times 1000}$$

where A = optical density at 665 and 649 nm, V = final volume (mL), and FW = leaf tissue fresh weight (g).

### 2.5. Measurements of Biochemical Indexes

The amount of MDA was determined by the thiobarbituric acid (TBA) reaction [33], and the absorbance of the reaction system was determined at 532 nm and 600 nm. $H_2O_2$ was determined by titanium sulfate colorimetry [34], and the absorbance of the reaction system was determined at 415 nm. The superoxide anion content was determined with the hydroxylamine oxygen method [35], and the absorbance of the reaction system was

determined at 530 nm. The content of free proline was determined by ninhydrin colorimetry [36], and the absorbance of the reaction system was determined at 520 nm. The levels of SOD were determined using the nitroblue tetrazolium method [37], and the absorbance of the reaction system was determined at 560 nm. The SOD enzyme activity in the reaction system was defined as one enzyme activity unit at 50% inhibition in the xanthine oxidase coupled reaction system. The amount of POD was determined by guaiacol colorimetry [38], and the absorbance of the reaction system was determined at 470 nm. The A470 change of 0.005 per minute per g of tissue in each mL reaction system is defined as an enzyme activity unit. All of the above were measured using an enzyme activity assay kit from Comin Biotechnology Co., Ltd., (Suzhou, China). All the physiological index measurements were repeated three times, and the statistical results were averaged.

*2.6. Statistical Analysis*

The experimental data obtained were processed in SPSS 25.0 with one-way ANOVA, Duncan's method for multiple comparisons ($p < 0.05$ significance level), and principal components analysis. All data showed normal distributions. Origin 2022 was used for creating graphs. The data were expressed as means $\pm$ SE. The comprehensive evaluation of salt tolerance in *P. talassica* $\times$ *P. euphratica* under different treatments was carried out using term weighting and membership function methods [39]. The membership function method formulas were as follows:

$$F(X_i) = \frac{X_i - X_{min}}{X_{max} - X_{min}} \tag{1}$$

$$F(X_i) = 1 - \frac{X_i - X_{min}}{X_{max} - X_{min}} \tag{2}$$

where F(Xi) represents the membership function value of each main factor, Xi represents the ith comprehensive index value of each variety, and Xmax and Xmin are the maximum and minimum values of the same index in each treatment, respectively. If there was a positive correlation between the measured index and salt tolerance, formula 1 was used; otherwise, formula 2 was used.

## 3. Results

*3.1. Effects of ABA, PP$_{333}$, and SA Application on the Whole Plant Dry Weight of P. talassica $\times$ P. euphratica under Salt Stress*

The 2% SSc treatment reduced the whole plant dry weight of *P. talassica* $\times$ *P. euphratica* seedlings (relative to the control) by 75.07% (Figure 1). After spraying exogenous ABA, PP$_{333}$, and SA (at different concentrations) onto the leaves (except for P1), the effect of salt stress on the dry weight of the seedlings was significantly alleviated, and the difference between those treatments using different concentrations of the same exogenous regulator reached a significant level. The dry weight, after using the 15 mg·L$^{-1}$ ABA, 900 mg·L$^{-1}$ PP$_{333}$, and 120 mg·L$^{-1}$ SA treatments, reached a maximum, which increased 2.57-, 2.35- and 2.59-fold, respectively, compared with that found for 2% SSc alone.

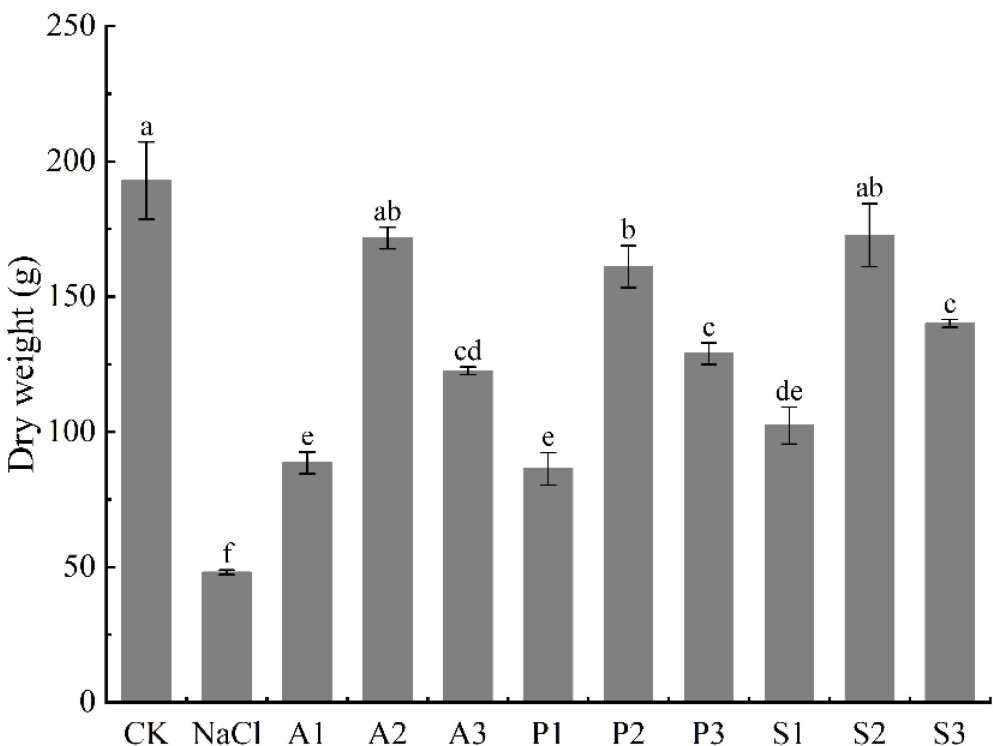

**Figure 1.** Effect of ABA, PP$_{333}$, and SA on the whole plant dry weight of *P. talassica* × *P. euphratica* under controlled and 2% SSc stress conditions. Different letters represent statistically significant differences ($p < 0.05$).

*3.2. Effects of ABA, PP$_{333}$, and SA Application on the Root Structure of P. talassica × P. euphratica under Salt Stress*

The 2% SSc treatment significantly reduced the root architecture of the *P. talassica* × *P. euphratica* seedlings relative to the control (Figure 2). After spraying the seedlings with different concentrations of exogenous ABA, PP$_{333}$, and SA, the effects of salt stress on the root architecture of *P. talassica* × *P. euphratica* was alleviated. With increasing ABA, PP$_{333}$, and SA concentrations, the root architecture of the seedlings showed a trend of first increasing and then decreasing.

The *P. talassica* × *P. euphratica* seedlings, when subjected to salt stress, showed significantly reduced total root length (77.06%), root surface area (72.57%), and root volume (68.93%) compared with the control (Figure 2). The reduction in root growth in the seedlings was alleviated by the exogenous application of ABA, PP$_{333}$, and SA. The reductions in root length were only 63.85, 29.06, and 46.67% following treatments A1, A2, and A3, respectively, 47.53, 19.27, and 35.09% following treatments P1, P2, and P3, respectively, and 46.14, 11.26, and 33.92% following treatments S1, S2, and S3, respectively, when compared to the control (Figure 2a). The reductions in root surface area were only 64.60, 28.29, and 46.72% following treatments A1, A2, and A3, respectively, 46.10, 12.89, and 40.39% following treatments P1, P2, and P3, respectively, and 36.65, 0.35, and 16.46% following treatments S1, S2, and S3, respectively, when compared to the control (Figure 2b). Finally, the reductions in root volume were only 54.27, 19.78, and 38.95% following treatments A1, A2, and A3, respectively, 44.36, 15.08, and 30.45% following treatments P1, P2, and P3, respectively, and 50.23, 6.18, and 25.79% following treatments S1, S2, and S3, respectively, when compared to the control (Figure 2c).

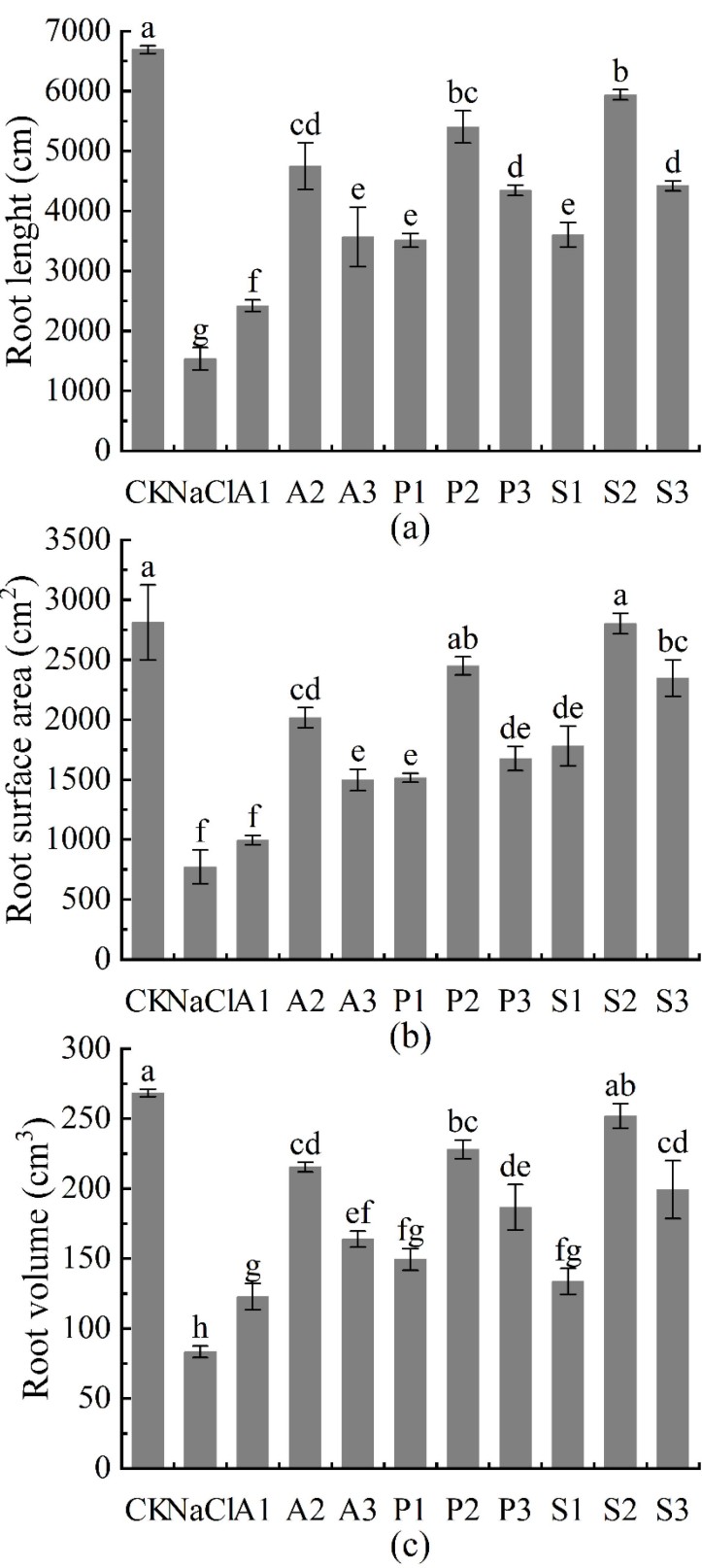

**Figure 2.** Effect of ABA, PP$_{333}$, and SA on the root length (**a**), root surface area (**b**), and root volume (**c**) of *P. talassica* × *P. euphratica* under control and 2% SSc stress conditions. Different letters represent statistically significant differences ($p < 0.05$).

*3.3. Effects of ABA, PP$_{333}$ and SA Application on Total Chl Content and Photosynthetic Variables of P. talassica × P. euphratica under Salt Stress*

After the addition of exogenous ABA, PP$_{333}$, and SA, the total Chl content of *P. talassica × P. euphratica* seedlings showed differences according to the type and concentration of exogenous hormones applied (Figure 3). The 2% SSc stress reduced the Chl content relative to the control by 65.01%. The three regulators produced similar trends in the total Chl contents, first increasing and then decreasing with an increase in hormone concentration. Spraying 15 mg·L$^{-1}$ ABA, 900 mg·L$^{-1}$ PP$_{333}$, and 120 mg·L$^{-1}$ SA had the best effect on relieving the stress induced by 2% SSc, resulting in only 7.66, 7.92, and 9.36% reductions in total Chl content, respectively. The rates of photosynthetic inhibition were 57.35, 57.09, and 55.65%, respectively, which was lower than under salt stress alone.

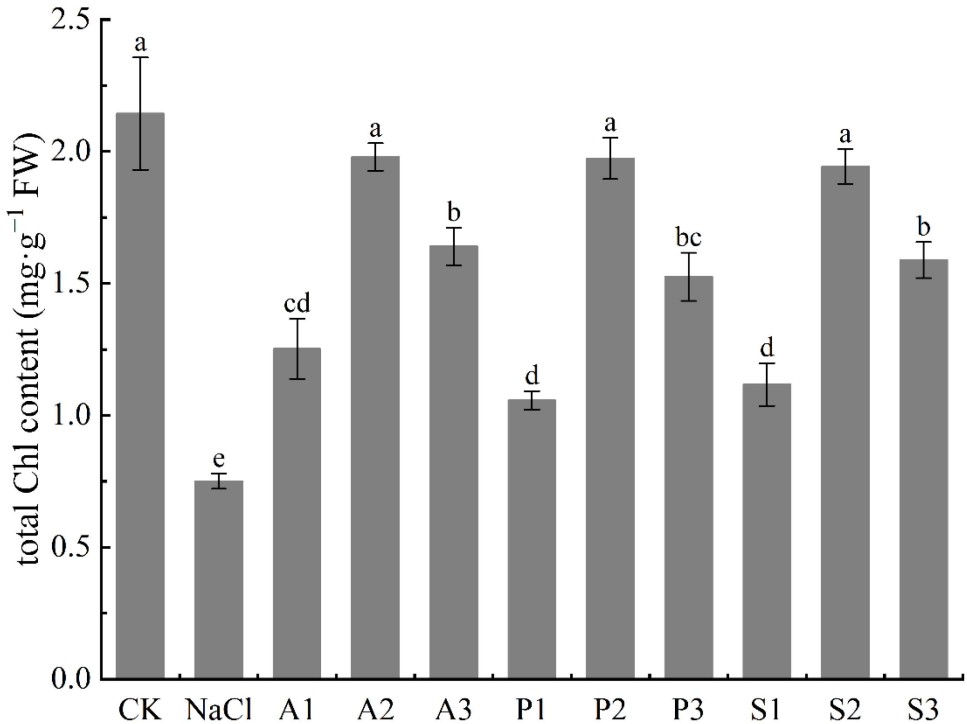

**Figure 3.** Effect of ABA, PP$_{333}$, and SA on total Chl content in *P. talassica × P. euphratica* under control and 2% SSc stress conditions. Different letters represent statistically significant differences ($p < 0.05$).

The Pn, Gs, and Tr directly reflected the differences in the photosynthetic characteristics of *P. talassica × P. euphratica* under various treatments (Figure 4). In this study, the Pn, Gs, and Tr of *P. talassica × P. euphratica* were significantly reduced under 2% SSc stress and were 43.94, 65.27, and 39.20% lower than the control, respectively. However, the Pn of the seedlings increased by 30.04, 63.10, and 35.39% following treatments A1, A2, and A3, respectively, 18.80, 73.03, and 43.80% following treatments P1, P2, and P3, respectively, and 26.69, 71.43, and 48.03% following treatments S1, S2, and S3, respectively, when compared with that observed under salt stress alone (Figure 4a). The increases in Gs were 33.36, 163.64, and 69.73% following treatments A1, A2, and A3, respectively, 78.82, 166.64, and 106.10% following treatments P1, P2, and P3, respectively, and 51.55, 178.82, and 127.28% following treatments S1, S2, and S3, respectively, when compared to salt stress alone (Figure 4b). The Tr values for *P. talassica × P. euphratica* increased by 20.00, 56.57, and 35.67% following treatments A1, A2, and A3, respectively; 13.43, 60.90, and 28.81% following treatments P1, P2, and P3, respectively, and 16.42, 57.61, and 37.61% following treatments S1, S2, and S3, respectively, when compared with salt stress alone (Figure 4c).

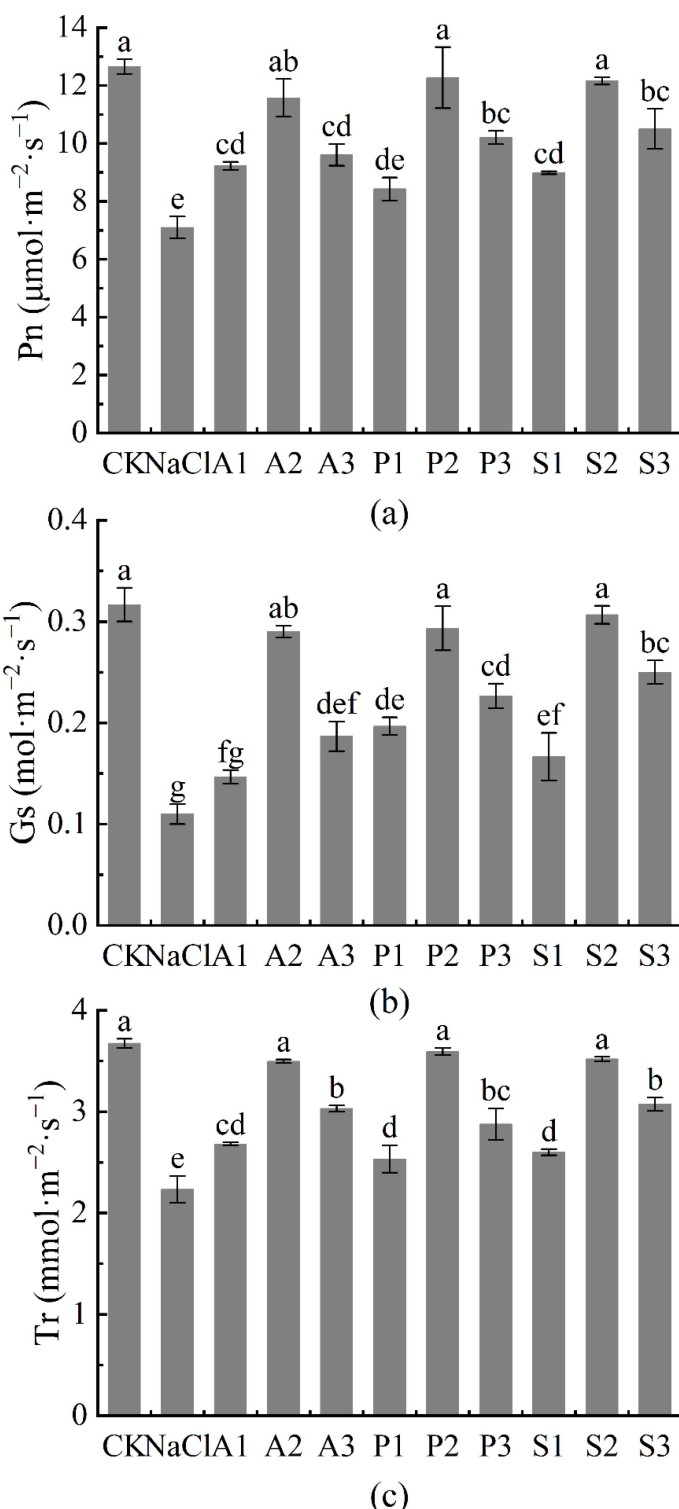

**Figure 4.** Effects of ABA, PP$_{333}$, and SA on Pn (**a**), Gs (**b**), and Tr (**c**) in *P. talassica* × *P. euphratica* under control and 2% SSc stress conditions. Different letters represent statistically significant differences ($p < 0.05$).

*3.4. Effects of ABA, PP$_{333}$, and SA Application on MDA, H$_2$O$_2$ and O$_2^-$ of P. talassica × P. euphratica under Salt Stress*

The 2% SSc treatment significantly increased the content of MDA, H$_2$O$_2$, and O$_2^-$ in the *P. talassica* × *P. euphratica* seedlings by 70.35, 86.35, and 33.50%, respectively, relative

to the control (Figure 5). The increased contents of MDA, $H_2O_2$, and $O_{2^-}$ were alleviated with the application of ABA, $PP_{333}$, and SA. The contents of MDA were 28.56, 41.93, and 29.09% higher in the control than those measured in seedlings under the A2, P2, and S3 treatments, respectively (Figure 5a). Moreover, under the A2, P2, and S2 treatments, the $H_2O_2$ and $O_2^-$ contents were significantly different from those under 2% SSc treatment, showing decreases of 42.09, 34.65, and 39.01% ($H_2O_2$ content) and 22.04, 23.18, and 22.45% ($O_2^-$ content), respectively (Figure 5b,c).

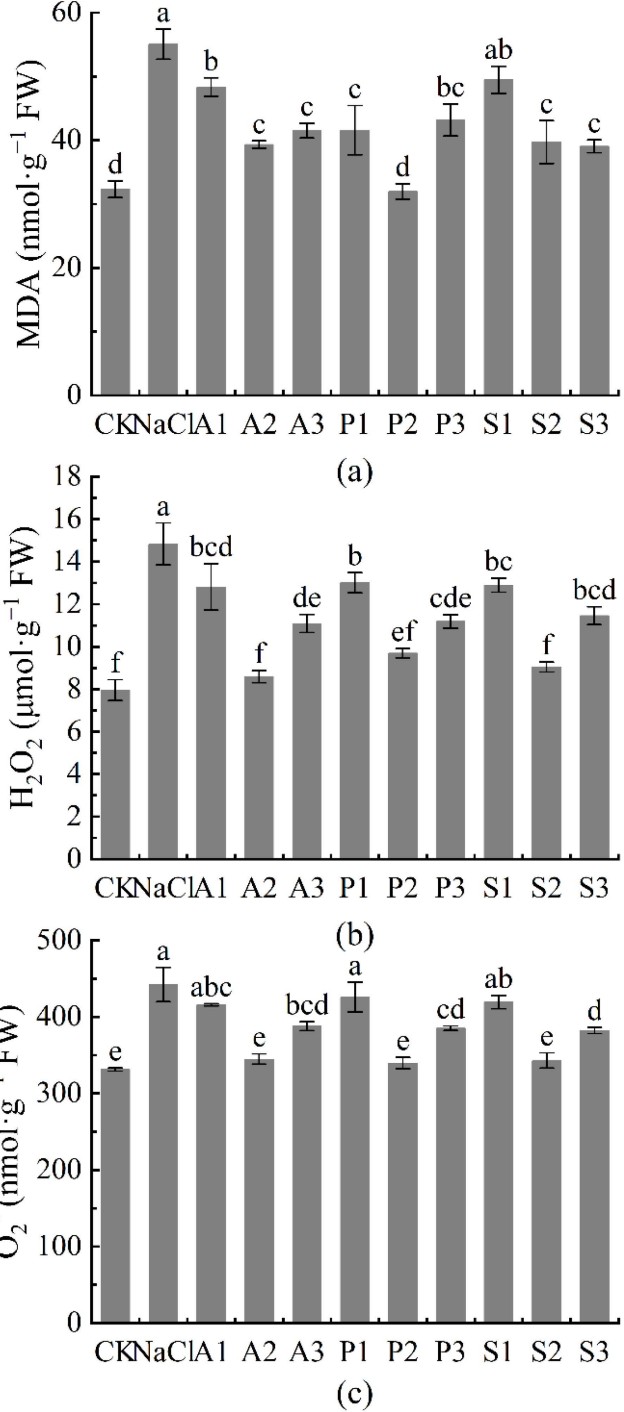

**Figure 5.** Effect of ABA, $PP_{333}$, and SA on MDA (**a**), $H_2O_2$ (**b**), and $O_2^-$ (**c**) in *P. talassica* × *P. euphratica* under control and 2% SSc stress conditions. Different letters represent statistically significant differences ($p < 0.05$).

### 3.5. Effects of ABA, PP$_{333}$, and SA Application on Pro of P. talassica × P. euphratica under Salt Stress

The 2% SSc stress increased the Pro content in *P. talassica* × *P. euphratica* (Figure 6). Compared with the control, the Pro content in the leaves of the seedlings increased by 31.10% under salt stress. Compared with salt stress, under treatments A1, A2, and A3, the Pro content increased by 10.00, 26.71, and 15.68%, respectively; treatments P1, P2, and P3 increased the Pro content by 12.40, 31.93, and 21.55%, respectively, and treatments S1, S2, and S3 increased the Pro content by 14.10, 37.15, 22.43%, respectively. Among them, the Pro content of the 120 mg·L$^{-1}$ SA treatment accumulated the most.

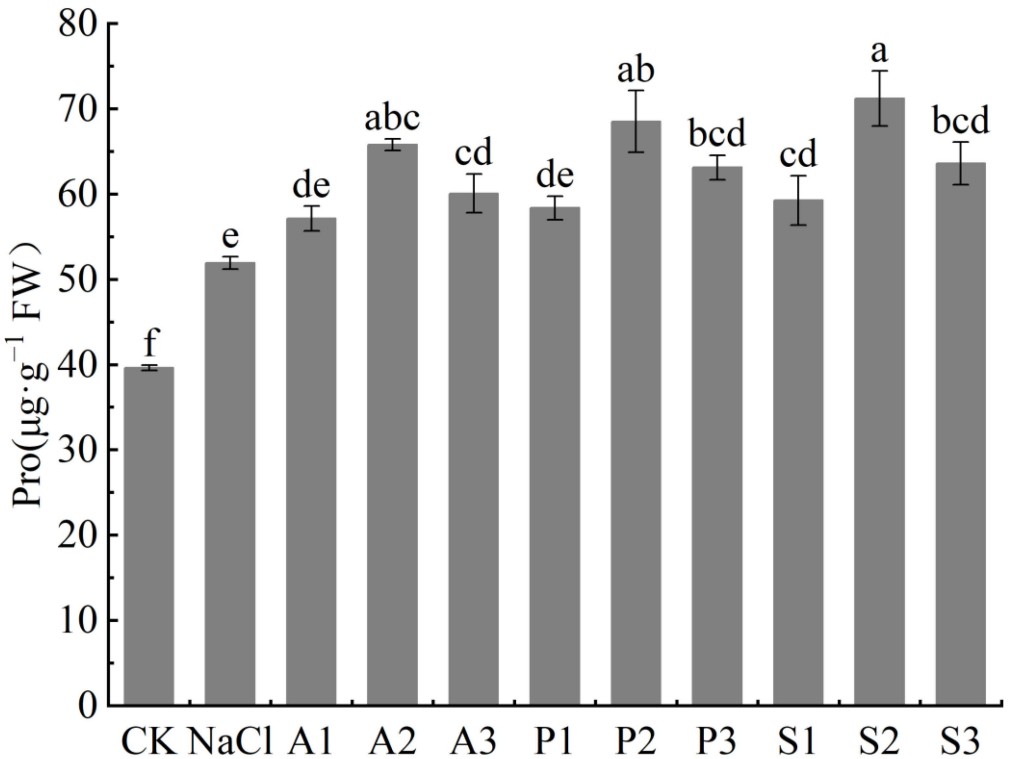

**Figure 6.** Effect of ABA, PP$_{333}$, and SA on Pro in *P. talassica* × *P. euphratica* under control and 2% SSc stress conditions. Different letters represent statistically significant differences ($p < 0.05$).

### 3.6. Effects of ABA, PP$_{333}$, and SA Application on Antioxidant Enzyme Activity of P. talassica × P. euphratica under Salt Stress

The 2% SSc stress increased the SOD and POD activities in *P. talassica* × *P. euphratica* (Figure 7). Compared with the control, the activities of SOD and POD in the leaves of the seedlings increased by 98.79 and 66.49%, respectively. Compared with salt stress, under treatments A1, A2, and A3, SOD activity increased by 10.54, 30.30, and 19.45%, and POD activity by 12.13, 46.79, and 33.03%, respectively; treatments P1, P2, and P3 increased SOD activity by 11.50, 33.48, and 21.77%, and POD activity by 6.24, 46.54, and 32.36%, respectively, and treatments S1, S2, and S3 increased SOD activity by 12.88, 34.06, and 23.47%, and POD activity by 26.05, 52.01, and 42.72%, respectively. The PP$_{333}$ treatment at 300 mg·L$^{-1}$ did not impact POD gene expression.

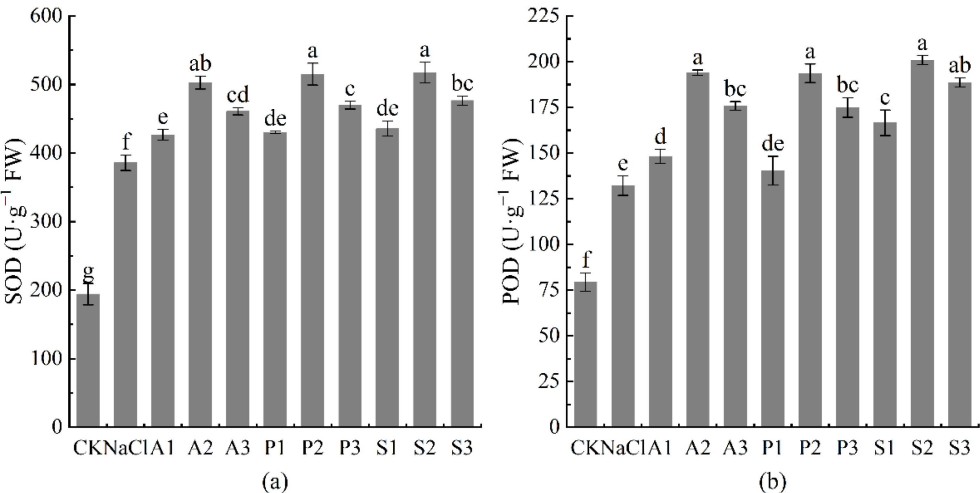

**Figure 7.** Effect of ABA, PP$_{333}$, and SA on SOD (**a**) and POD (**b**) in *P. talassica* × *P. euphratica* under control and 2% SSc stress conditions. Different letters represent statistically significant differences (*p* < 0.05).

*3.7. Comprehensive Evaluation and Analysis of the Effects of the Three Plant Growth Regulators on the Growth, Photosynthesis, Physiological, and Biochemical Indexes of P. talassica × P. euphratica under Salt Stress*

To avoid the limitation of a single index evaluation, a comprehensive evaluation of 14 indexes, including the dry weight, root length, and root surface area of the *P. talassica* × *P. euphratica* seedlings under the treatment of three plant growth regulators, were measured to judge the impact of different external regulators on the salt tolerance of *P. talassica* × *P. euphratica* under 2% SSc stress (Table 2). The order of evaluation for the different treatments was S2 > P2 > A2 > S3 > P3 > A3 > S1 > P1 > A1. In summary, 120 mg·L$^{-1}$ SA had the best effect on enhancing the salt tolerance of the *P. talassica* × *P. euphratica* seedlings.

**Table 2.** Comprehensive evaluation of the salt tolerance of *P. talassica* × *P. euphratica* after spraying seedlings with different exogenous regulators.

| Index | Membership Function Value | | | | | | | | | | | Weight of Indicators % |
|---|---|---|---|---|---|---|---|---|---|---|---|---|
| | CK | NaCl | A1 | A2 | A3 | P1 | P2 | P3 | S1 | S2 | S3 | |
| Dry weight | 0.88 | 0.01 | 0.25 | 0.75 | 0.45 | 0.24 | 0.69 | 0.49 | 0.33 | **0.76** | 0.56 | 8.96% |
| Root length | 0.99 | 0.05 | 0.21 | 0.63 | 0.42 | 0.41 | 0.75 | 0.56 | 0.43 | **0.85** | 0.57 | 8.70% |
| Root surface area | 0.79 | 0.07 | 0.15 | 0.51 | 0.33 | 0.33 | 0.66 | 0.39 | 0.43 | **0.79** | 0.63 | 8.37% |
| Root volume | 0.98 | 0.03 | 0.23 | 0.71 | 0.44 | 0.37 | 0.77 | 0.56 | 0.29 | **0.89** | 0.62 | 8.93% |
| Chl | 0.78 | 0.03 | 0.30 | 0.69 | 0.51 | 0.19 | 0.69 | 0.45 | 0.23 | **0.67** | 0.48 | 8.82% |
| Pn | 0.84 | 0.09 | 0.38 | 0.69 | 0.43 | 0.27 | 0.79 | 0.51 | 0.34 | **0.77** | 0.55 | 8.92% |
| Gs | 0.87 | 0.07 | 0.21 | 0.77 | 0.38 | 0.40 | 0.78 | 0.53 | 0.29 | **0.83** | 0.61 | 8.90% |
| Tr | 0.95 | 0.11 | 0.37 | 0.85 | 0.58 | 0.28 | 0.90 | 0.48 | 0.32 | **0.86** | 0.60 | 8.88% |
| MDA | 0.91 | 0.16 | 0.38 | 0.68 | 0.61 | 0.61 | 0.93 | 0.55 | 0.34 | **0.67** | 0.69 | 8.19% |
| H$_2$O$_2$ | 0.92 | 0.20 | 0.41 | 0.85 | 0.59 | 0.39 | 0.74 | 0.58 | 0.40 | **0.80** | 0.55 | 8.80% |
| O$_2^-$ | 0.97 | 0.27 | 0.44 | 0.89 | 0.61 | 0.37 | 0.92 | 0.63 | 0.41 | **0.90** | 0.65 | 8.89% |
| Pro | 0.02 | 0.35 | 0.49 | 0.72 | 0.57 | 0.52 | 0.80 | 0.65 | 0.55 | **0.87** | 0.66 | 1.86% |
| SOD | 0.08 | 0.58 | 0.69 | 0.89 | 0.78 | 0.70 | 0.92 | 0.80 | 0.71 | **0.93** | 0.82 | 0.06% |
| POD | 0.05 | 0.45 | 0.57 | 0.91 | 0.77 | 0.51 | 0.91 | 0.77 | 0.71 | **0.96** | 0.87 | 1.73% |
| Weighted average | 0.87 | 0.11 | 0.31 | 0.73 | 0.49 | 0.36 | 0.79 | 0.53 | 0.36 | **0.80** | 0.60 | |
| Ranking | 1 | 11 | 10 | 4 | 7 | 8 | 3 | 6 | 8 | **2** | 5 | |

## 4. Discussion

Proper soil salinity can promote the growth of many halophytes [40]. However, excessive soil salinity will cause salt damage to plants and inhibit their growth. Therefore, plants must establish a variety of regulatory mechanisms to respond to salt stress and maintain growth [41]. The rational use of plant growth regulators in the form of foliar sprays can mitigate the adverse effects of salt stress [42]. Research shows that ABA can improve photosynthetic capacity and biomass accumulation and alleviate the physiological damage caused by stress to plants [43]. Paclobutrazol can improve the stress resistance of *Sequoia sempervirens* seedlings by improving their photosynthetic characteristics [44]. SA can improve the antioxidant capacity and salt tolerance of potato [45]. This study clarified the physiological response mechanism of spraying plant growth regulators onto *P. talassica × P. euphratica* under soil salt stress, which is necessary to alleviate soil salt stress and improve the utilization of saline soils in forestry.

In this study, we evaluated the effects of soil salinity and plant growth regulators on the growth indicators of *P. talassica × P. euphratica*. The whole plant biomass and root parameters of *P. talassica × P. euphratica* were significantly reduced at 2% SSc treatment. The osmotic stress and ion toxicity of soil salinity were the main reasons for the growth decline. Roots provide fixation and support for plants while absorbing nutrients and water from the soil and transmitting it to supply its growth and development needs [46]. Its morphological structure reflects the degree of development of the plant root system and also the growth state of the whole plant. Therefore, when the salt content of soil increases, the root architecture is inhibited by the stress, which restricts the growth and development of plants and even leads to their death [47]. However, when foliar spraying ABA, $PP_{333}$, and SA onto *P. talassica × P. euphratica*, its biomass, total root length, root surface area, and root volume showed a trend of first increasing and then decreasing with increasing regulator concentrations. Both 900 mg·$L^{-1}$ $PP_{333}$ and 120 mg·$L^{-1}$ SA had the most obvious positive regulatory influence on the root parameters of the *P. talassica × P. euphratica* seedlings and the best mitigating effects. Plant growth regulators played a great role in restoring the salt resistance of the *P. talassica × P. euphratica* roots, making them stronger, enhancing their ability to absorb nutrients and water, and alleviating the stressful effects of the salt environment on *P. talassica × P. euphratica* to some extent.

Chl enables plant cells to absorb and convert light energy and is the main pigment involved in photosynthesis. Its content can be used as a parameter for physiological metabolism, nutritional status, and aging in the leaves [48]. Researchers have discovered that SA stimulates the synthesis of Chl and slows the rate of Chl reduction, thus delaying a decline in Chl content [49,50]. Our investigation demonstrated a reduction in total Chl content in the leaves of *P. talassica × P. euphratica* under salt stress and a decline in its ability to assimilate light energy, which may be related to the enhancement of Chl enzyme activity under salt stress to promote Chl decomposition. The electron transport chain and energy transport in the photosynthetic system may also be impacted [51], causing the inhibition of photosynthesis. In addition, salt stress can inhibit Pn, Tr, and Gs in the leaves of *P. talassica × P. euphratica*, thereby reducing photosynthetic efficiency, which is similar to the results for *Ulmus pumila* [52] and *Populus cathayana* [53]. This may occur because salt stress leads to a decrease in Gs in the leaves, making it difficult to supply the $CO_2$ concentration required for photosynthesis and leading to a decrease in the rate of photosynthesis. After spraying exogenous ABA, $PP_{333}$, and SA on the leaves of *P. talassica × P. euphratica*, the Pn, Tr, and Gs increased first and then decreased with increasing hormone concentration, but all increased to varying degrees. Foliar sprays of 15 mg·$L^{-1}$ ABA, 900 mg·$L^{-1}$ $PP_{333}$, and 120 mg·$L^{-1}$ SA had the most obvious positive regulatory influence on the photosynthetic parameters of *P. talassica × P. euphratica* seedlings and the best mitigating effects. Spraying exogenous hormones on the leaves can adjust the Gs, accelerate carbon carboxylation, slow stomatal restriction, and enhance the photosynthetic capacity of plants [54,55]. ABA, $PP_{333}$, and SA have been shown to alleviate the adverse effects of abiotic stress on plants and

improve photosynthetic performance [56]. The results found in this study were similar to those reported for *Populus* [57], *Curcuma longa* [58], and *Perennial Ryegrass* [59].

MDA is one of the main products of membrane lipid peroxidation. It is combined with $H_2O_2$ and $O_2^-$, which are a measure of oxidative damage to plants. Excessive reactive oxygen species and MDA lead to membrane peroxidation, increased membrane permeability, and damage to the plant's defense systems, affecting physiological and biochemical metabolism [60]. In this study, the contents of MDA, $H_2O_2$, and $O_2^-$ in *P. talassica* × *P. euphratica* seedlings under salt stress increased significantly, which damaged the cell membrane system and led to the continuous accumulation of membrane lipid peroxides. The spraying of ABA, $PP_{333}$, and SA can significantly reduce the content of MDA, $H_2O_2$, and $O_2^-$, and alleviate the damage of membrane lipid peroxide, which is induced by salt injuries to plants. Pro can regulate the osmotic potential of plant cells. The accumulation of its content can enhance osmotic adjustment ability, reduce the inhibition of antioxidant enzyme activity under salt stress, and, thus, improve the salt tolerance of plants [61]. In this study, Pro content tended to increase under salt stress, indicating that salt stress promotes the accumulation of proline to mitigate salt damage. ABA, $PP_{333}$, and SA treatments increased the content of proline, indicating that they had the function of regulating proline metabolism and could alleviate the damage of the *P. talassica* × *P. euphratica* seedlings under salt stress. The results found in this study were similar to those reported for rice seedlings [62] and *Mentha simplex* [63].

SOD and POD are two important antioxidant enzymes for scavenging active oxygen from plants. Improving the activity of these enzymes can alleviate growth inhibition under salt stress and improve the resistance of the plant seedlings. SOD can catalyze the conversion of $O_2^-$ to $H_2O_2$ [64], and POD can convert $H_2O_2$ to $H_2O$. Exogenous ABA, $PP_{333}$, and SA enhanced the activity of SOD and POD in the leaves of *P. talassica* × *P. euphratica* and reduced the increase in MDA, $H_2O_2$, and $O_2^-$ contents under salt stress, indicating that these three exogenous plant growth regulators can alleviate the oxidative stress caused by abiotic stress. ABA is involved in the restoration of osmotic and antioxidant mechanisms of rice under salt stress [65]. $PP_{333}$ can improve the antioxidant capacity of sweet sorghum under salt stress [66]. This is similar to the results obtained in *Phoenix dactylifera* [67], *Punica granatum* [68], and *Platycladus orientalis* [69].

In conclusion, our results showed that the foliar spraying of exogenous ABA, $PP_{333}$, and SA at a certain concentration could adjust the osmotic capacity of *P. talassica* × *P. euphratica* seedlings, reduce the degree of membrane lipid peroxidation, enhance cell membrane stability and the antioxidant defense system, promote root growth and development and biomass accumulation, reduce chlorophyll decomposition, and improve photosynthetic performance, thereby reducing the adverse effects of salt stress on *P. talassica* × *P. euphratica* seedlings. However, how exogenous ABA, $PP_{333}$, and SA affect the molecular mechanisms of salt resistance and endogenous hormone interactions on *P. talassica* × *P. euphratica* seedlings remains to be further researched in the future.

## 5. Conclusions

Our results revealed that 2% SSc stress significantly inhibited the growth, root structure, photosynthetic characteristics, and physiological and biochemical characteristics of *P. talassica* × *P. euphratica*. Foliar spraying with ABA, $PP_{333}$, and SA in appropriate concentrations significantly promoted root morphological development, which was conducive to the accumulation of dry matter in the plants, alleviating the damage caused by salt stress and promoting the growth of leaves and, thus, increasing chlorophyll content and photosynthetic capacity. Meanwhile, ABA, $PP_{333}$, and SA could inhibit increases in MDA, $H_2O_2$, $O_2^-$, and proline contents, protecting the stability of the cell membrane structure and improving the activity of SOD and POD antioxidant enzymes, which could increase cold resistance and reduce the production of reactive oxygen species. The optimal concentrations of ABA, $PP_{333}$, and SA for alleviating salt stress were 15 mg·L$^{-1}$, 900 mg·L$^{-1}$, and 120 mg·L$^{-1}$, respectively. Among them, 120 mg·L$^{-1}$ SA was the most effective for protecting *P. talassica* × *P. euphratica* against salt stress.

**Author Contributions:** Z.H. conceived and directed the project; M.S. and M.Z. performed all the experiments and the statistical analyses, and wrote the paper; Y.L. participated in part of the experiments. All authors have read and agreed to the published version of the manuscript.

**Funding:** This work was funded by the Open Project of Xinjiang Production and Construction Corps Key Laboratory of Protection and Utilization of Biological Resources in Tarim Basin (Funding number: BRZD1902, Funder: Corps Key Laboratory). This work was also funded by Selection and Cultivation Project of "Talents of Xinjiang Production and Construction Corps" (Funding number: 38000020924, Funder: Xinjiang Production and Construction Corps).

**Institutional Review Board Statement:** Not applicable.

**Informed Consent Statement:** Not applicable.

**Data Availability Statement:** Not applicable.

**Conflicts of Interest:** The authors declare that they have no known competing financial interest or personal relationships that could have appeared to influence the work reported in this paper.

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
