# Peer review of "Abscisic Acid, Paclobutrazol, and Salicylic Acid Alleviate Salt Stress in Populus talassica × Populus euphratica by Modulating Plant Root Architecture, Photosynthesis, and the Antioxidant Defense System"

_forests, doi:10.3390/f13111864_

Round 1

Reviewer 1 Report

In this study, the authors evaluated the effect of Abscisic Acid, Paclobutrazol and Salicylic Acid Alleviate Salt Stress in Populus talassica × Populus euphratica by Modulating Plant Root Architecture, Photosynthesis and the Antioxidant Defense System.

In my opinion, the manuscript needs several improvements before accepting it for publication.

Abstract

Introduce a short sentence on the problematic of the present study.

-- L18: Replace "inhibited" by "decreased" and ovoid "decreased" in line 19

-- L20-25: Add “compared to the control under 2% NaCl stress” in “However, the exogenous application of ABA ….. and superoxide anion (O2) compared to the control under 2% NaCl stress”

Please provide a more appropriate conclusion highlighting the importance of these data in combating soil salinization and increasing agricultural growth and yield in saline soils.

Keywords

Please indicate other keywords that are not included in the title to avoid repetition.

Introduction

-- L38-39: Please add references with more recent statistics and avoid older statistics (2010).

-- L44-45: “Xinjiang … sustainable forestry development (Add references)”.

-- L50 : Add recent references such as "Ait-El-Mokhtar, Mohamed, et al. "Use of mycorrhizal fungi in improving tolerance of the date palm (Phoenix dactylifera L.) seedlings to salt stress." Scientia Horticulturae 253 (2019): 429-438" et "Ben Laouane, R., et al. "Effects of arbuscular mycorrhizal fungi and rhizobia symbiosis on the tolerance of Medicago sativa to salt stress." Gesunde Pflanzen 71.2 (2019): 135-146".

-- L70-71: Add recent references “…. the physiological damage caused”.

-- L71-72: Please add other recent references

-- L92-93: Add recent references “P. talassica × P. euphratica, an … the male parent”.

-- L94-96: Add recent references “In addition, it has …. areas of northwest China”.

Materials and Methods

L116-117: Please mention the method of application of salt stress (2%) in detail and add the reference if possible.

L120-125: If possible, please add treatments with the same hormones without salt stress.

Please indicate on what basis the selection of P. talassica × P. euphratica seedlings and also 2% NaCl was made.

If it is possible to add other characteristics of the soil used such as electrical conductivity, the contents of N, P, K, Ca and Na, which are very important in this study.

Replace “H2O2 and O2-“ by “H2O2 and O2-“ in all the text

Please add the climatic conditions in which the culture takes place "temperature, light intensity, humidity....".

L130: The number of repetitions (three) is very limited, you must perform at least five repetitions for the results to be more reliable.

L151: Please indicate the protocol used for the measurements of the biochemical indices (MDA, H2O2, O2-, SOD and POD) in detail.

L152: Please write correctly "H2O2, O2-"

L159 Statistical Analysis

The test you used does not fit your data. indeed, your data are very complex because they contain different factors :

-       The application of different concentrations of ABA, PP333 and SA (treatments)

-       The salt stress conditions

It means that your data should be analyzed with a two-way ANOVA, which is very complex ! I think that a two-way ANOVA (treatment x stress) could be simplier and relatively well accepted by reviewers. I advise you to contact a statistician to analyze your data, because a simple ANOVA is not relevant in your study.

L182-185: this sentence should be more appropriated in the discussion part, only explain significant differences in the results part.

L175: Please indicate the corresponding dry weight (shoot or root?)

L200-207: How do you explain that after 45 days of culture, this difference in root length between the different treatments and the NaCl treatment. Please check these values

Same remark for the surface and the volume of the root. Please check these values.

L230-234 this sentence should be more appropriated in the discussion part, only explain significant differences in the results part.

In the results section, you should just interpret the data without making any conclusions. Please delete inappropriate phrases in the results section.

Discussion

L316-325: A multitude of generalities, please restate the purpose of your study.

L327: Replace "for plant growth" by "for their growth" to avoid repeating plants twice.

In this section, it is necessary to mention the potential mechanisms of the hormones applied in root growth and architecture and it is also essential to indicate the effect of the concentration for each parameter measured.

Conclusions

Éviter de répéter les résultats obtenus et indiquer également la pertinence de cette étude en termes de capacité de ces hormones à améliorer la productivité agricole et la possibilité de les utiliser sur le terrain.

Reference

-       Authors paid attention to the quality of the references and the list, thank you.

-       Please pay attention to the order of references in the main text.

Reviewer 2 Report

Journal: Forests

Manuscript ID: forests-1979882

The manuscript entitled:”Abscisic Acid, Paclobutrazol and Salicylic Acid Alleviate Salt Stress in Populus talassica × Populus euphratica by Modulating Plant Root Architecture, Photosynthesis and the Antioxidant Defense System“, by Meland et al. falls perfectly within the scope of Horticulturae journal.

The ideal doses of these exogenous plant growth regulators were explored in this work for P. talassica x P. euphratica seedlings under salt stress to elucidate the regulatory mechanism of salt tolerance and to offer a reference and theoretical foundation for the use of plant growth regulators in easing growth inhibition in salt stressed seedlings. As the study is very important, I recommend considering the following revisions.

·         Is there any practical guide of the exogenous application of Plant Growth Regulators (PGRs) in the literature. Please add more references for the section lines 65-69.

·         In subsection Plant Materials and Experimental Treatments, consider formatting the treatments (experimental design) in a tabular form, for a better visualization.

·         Subsection, Measurement of Growth Index, Can the authors describe more their random sampling method (RSM), as there is many types of RSM (Stratified, Cluster, systematic…).

·         The formula used in the calculation of total chlorophyl must be inserted as an equation using the appropriate function (in word), consider writing the absorbances as indices (A649; A665); same for the membership function method formulas (lines 165-167).

·         Use bold to highlight the best results in Table 1. (for a better visualization by the reader).

·         In such type of studies, the quantification of proline is primordial. Proline is a known osmoprotectant amin acid that plays a crucial role in osmotic balancing, the protection of the sub-cellular compartments, structures, and enzymes, and in increasing turgor pressure when needed following a complex mechanism of action. As such, why did the authors “neglect” the quantification of proline and limited their work in studying enzymatic mechanisms of action only!

·         What are the perspectives of this study, the exogenous application of SA in such a high dose (120 mg. L-1), may alter other physiological and biochemical patterns in the studied plant. Discovering these changes will make the use of SA exogenously, reasonable.

Minor corrections:

- Consider rewriting the following words, chemical formulas appropriately, H2O2; O2- (lines 127-128, 152), CO2 (line 141), check throughout the text.

- Conider writing plant’s scientific names in Italics.

Round 2

Reviewer 2 Report

The authors have carefully revised the manuscript according to my suggestions. Thus, I recommend the publication at this current form.